# Transcriptome Level Analysis of Genes of Exogenous Ethylene Applied under Phosphorus Stress in Chinese Fir

**DOI:** 10.3390/plants11152036

**Published:** 2022-08-04

**Authors:** Shuotian Huang, Lixia Zhang, Tingting Cai, Yuxuan Zhao, Jiao Liu, Pengfei Wu, Xiangqing Ma, Peng Shuai

**Affiliations:** 1College of Forestry, Fujian Agriculture and Forestry University, Fuzhou 350002, China; 2Chinese Fir Engineering Technology Research Center of the State Forestry and Grassland Administration, Fuzhou 350002, China

**Keywords:** *Cunninghamia lanceolata*, UMI RNA-seq, phosphorus treatment, ethephon, transcription factor

## Abstract

Chinese fir (*Cunninghamia lanceolata* (Lamb.) Hook) is a widely grown gymnosperm in China. Phosphorus (P) is an indispensable nutrient for the growth of Chinese fir. Inorganic phosphate (Pi) deficiency exists in soils of many Chinese fir planting area regions, and the trees themselves have limited efficiency in utilizing P from the soil. Ethylene is important in regulation responses to nutrient deficiencies. However, little is known about how ethylene signals participate in Pi stress in Chinese fir. A total of six different treatments were performed to reveal the transcript levels of Chinese fir under Pi, ethephon (an ethylene-releasing compound), and CoCl_2_ (cobalt chloride, an ethylene biosynthesis inhibitor) treatments. We assembled a full-length reference transcriptome containing 22,243 unigenes as a reference for UMI RNA-seq (Digital RNA-seq). There were 586 Differentially Expressed Genes (DEGs) in the Pi starvation (NP) group, while DEGs from additional ethephon or CoCl_2_ in NP were 708 and 292, respectively. Among the DEGs in each treatment, there were 83 TFs in these treatment groups. MYB (v-myb avian myeloblastosis viral oncogene homolog) family was the most abundant transcription factors (TFs). Three ERF (Ethylene response factor) family genes were identified when only ethylene content was imposed as a variable. Enrichment analysis indicated that the ascorbate and aldarate metabolism pathway plays a key role in resistance to Pi deficiency. This study provides insights for further elucidating the regulatory mechanism of Pi deficiency in Chinese fir.

## 1. Introduction

Chinese fir (*Cunninghamia lanceolata* (Lamb.) Hook) is a coniferous species native to southern China and is widely planted in China due to its fast-growing and suitable wood performance properties [1]. Continuous planting of Chinese fir can lead to a decrease in soil orthophosphate content, which in turn affects its productivity [2]. Phosphorus (P) is one of the indispensable nutrients for plants in the growth process. Inorganic phosphate (Pi) is combined with metal cations to form insoluble P complexes in the soil, and organic phosphate in the soil also needs to be enzymatically phosphorylated before it can be absorbed by the roots, so the level of P available to plants is limited, and only free Pi is available in the soil solution [3,4]. Research on how to improve plant nutrient use efficiency is a potential way to address the low availability of soil P [5].

Plants adapt to Pi deficiency by a series of internal response mechanisms that are physiologically, morphologically, and biochemically manifested, and these adaptations are collectively referred to as the Pi Starvation Response (PSR), which prioritizes Pi use and ensures maximum external Pi use by plants [4]. Chinese fir showed a significant decrease in Pi deficiency and an increase in organic acid content in seedling roots under Pi deficiency. Its genes related to Pi starvation were expressed, citric acid and malic acid synthesis genes were increased, and its Pi metabolic pathways were complex and diverse [6,7]. Hormones play a key role in Pi deficiency, and hormones such as ethylene, Auxin, and abscisic acid (ABA) have been reported [8,9,10]. Some hormones regulate Pi deficiency positively, while others regulate Pi deficiency inversely, and hormones regulate Pi deficiency not only individually but also by coordinating Pi deficiency through similar signals involved in both responses [11,12]. Ethylene plays an important role in the adaptive mechanism of plant stress resistance. Low P conditions can induce ethylene production and positively regulate the Pi starvation pathway. Ethylene can regulate changes in root structure under Pi deficiency, as in *Arabidopsis thaliana*, the primary root growth rate is reduced, and lateral root density is increased, and these can be mitigated with ethylene inhibitors [13,14]. Ethephon is a commonly used growth regulator. In *Cucurbita moschata,* ethephon affects the synthesis of ethylene and the transcription of signaling genes, initiating the maturation process in the plant and also causing changes in the endogenous hormone levels in the shoot apex meristematic tissue [15]. CoCl_2_ (cobalt chloride) is an ethylene synthesis inhibitor, and cobalt ions can inhibit ethylene production and action by blocking specific steps in biosynthesis or signaling pathways, respectively [16]. Under P stress conditions, low concentrations of CoCl_2_ promoted both POD and SOD activities in Chinese fir seedlings [17]. The application of CoCl_2_ in desert grasslands can extend the life span of plant leaves by reducing ethylene content [18]. Therefore, under Pi deficiency, we used ethephon and CoCl_2_ to control the exogenous ethylene content separately to observe its response in the genes. In addition, *Oryza sativa* ethylene signaling can enhance acid phosphatase activity by regulating cell wall pectin and enhancing P translocation into the shoot [19]. *Arabidopsis Pi* starvation response leads to anthocyanin accumulation induced by overexpression of AP2/ERF (APETALA2/ethylene responsive factor) genes [20]. Some microRNA can inhibit Pi starvation signaling from the ground to the root, thus interacting positively with ethylene in the activation response. This has been extensively studied in different plants [21,22,23,24]. It has also been verified in different *Arabidopsis* ethylene gene mutants that Pi deficiency can cause changes in phosphatase activity [25]. In summary, ethylene is involved in various additional plant adaptations to P limitation, including inter-root P mobilization, P uptake, and internal P cycling, but there is considerable genetic variation among plant species and varieties, and more specific and focused research is needed. Therefore, Chinese fir seedlings were chosen for the experiment to explore the similarities and differences with other species under P stress and exogenous ethylene treatment.

Currently, most of the research on plant ethylene response to Pi deficiency is still focused on the model plant *Arabidopsis*, which contributes to our understanding of the P signaling response pathway, but this understanding is still very limited [20,26,27]. Physiologically, Chinese fir can respond to Pi starvation with exogenous ethylene to improve the tolerance of seedlings to Pi deficiency [17]. At the transcriptome level, there have been partial articles on RNA-seq for P stress in Chinese fir, but RNA-seq from the perspective of studying P response from the application of exogenous ethylene has not been implemented [7,28,29]. We mined Chinese fir with more consistent growth by RNA-seq for differential genes and transcription factors in Chinese fir regarding the co-interaction of Pi and ethylene. On this basis, we proposed the following hypotheses: (1) some genes respond together when exogenous ethylene was applied under P stress and only under P stress (2) these response genes may respond under these conditions: when only ethephon was added, when CoCl_2_ was added, and when both substances were added. These results provide genetic information related to P stress and ethylene, which can provide a basis for further development and evaluation of integrated stress-resistant Chinese fir.

## 2. Results

### 2.1. Chlorophyll and Anthocyanin Contents of Chinese Fir Seedlings

As shown in (Figure 1A), the chlorophyll content accumulated significantly in the case of Pi deficiency compared to normal Pi supply, but the changes in chlorophyll content became insignificant after the application of ethephon and CoCl_2_, respectively. As shown in (Figure 1B), the changes in anthocyanin were insignificant in the case of Pi deficiency compared to normal Pi supply and were also insignificant after the application of ethephon, but the differences in Pi became significant after the application of CoCl_2_.

### 2.2. Transcriptome Dataset of Chinese Fir Roots

We sequenced a root, stem, and leaf mix using the PacBio Sequel II platform to obtain the full-length reference transcriptome of Chinese fir. A total of 71.32G of subreads base was generated. The polished consensus was clustered and de-redundant using CD-HIT-EST software based on 95% similarity between sequences to obtain 22,243 unigenes with an average length of 2079 bp, respectively (Table 1). A total of 22,243 unigenes were annotated by BLAST, and the specific numbers of unigenes were annotated in the seven databases in Appendix A. The number of transcription factor annotations in Appendix A.

Twenty-four UMI RNA-seq libraries were constructed on the Illumina platform to reveal gene expression profiles of Pi deficiency stress and exogenous ethylene Chinese fir roots. These libraries were used to correct full-length transcriptome data as well as to obtain expression under different treatments. High-quality paired-end reads were generated after screening sequences with low base quality (Appendix A). The overall sequencing error rate was 0.03%, indicating that the sequencing quality was reliable and ready for the next step of the analysis.

### 2.3. Response of DEGs under P Treatment with Application of Ethephon or CoCl_2_

#### 2.3.1. Identification of DEGs under Relevant Treatments

CKP (normal Pi treatment) was used as a control and compared with NP (Pi starvation treatment), NP_ETH (Pi starvation and ethephon treatment), and NP_Co (Pi starvation and CoCl_2_ treatment). The differential genes for Pi deficiency, external ethylene addition under Pi deficiency, and external ethylene inhibition under Pi deficiency could be compared. A total of 586, 708, and 292 DEGs were found for NP, NP_ETH, and NP_Co, respectively (Figure 2A), with 35 genes responding together and 357, 467, and 133 genes responding alone, respectively (Figure 2A). The presence of more DEGs in the ethephon treatment group than in the CoCl_2_ treatment group. The TFs from DEGs were also counted, and a total of 83 TFs were compared. Each group of treatments had some TFs involved in the response. The highest number of TFs on the annotation of DEGs in the NP_ETH group was 46 (Appendix A). However, in terms of the percentage, the NP group TFs accounted for the largest proportion from the DEGs.

Three DEGs were significantly expressed in each treatment, namely transcript13260/f5p0/2398 (ABC transporter), transcript11003/f13p0/2395 (serine-type endurance activity), protein kinase of transcript21015/f3p0/2069 (oxidation-reduction process), (Table 2). When ethephon was added, it promoted the negative regulation of Pi deficiency, while CoCl_2_ seemed to alleviate the negative regulation of Pi deficiency.

#### 2.3.2. Functional Enrichment of DEGs

Gene ontology (GO) enrichment includes three major functional sets, cellular component (CC), molecular function (MF), and biological process (BP), and our treatment group showed outstanding performance in MF, and BP, as shown in Appendix A. When the CKP group was used as a control, all three groups were enriched in serine-type enzyme activity, which was significantly enriched in NP_ETH. The NP_ETH was enriched with a few DEGs of chlorophyll metabolic process, chlorophyll biosynthetic process, and enriched DEGs of transport and ion binding functions. In NP_Co, there were two phosphate ion transport enriched, enriched to catalytic activity, ion binding, oxidoreductase activity, and oxidation-reduction process function of DEGs enriched. Kyoto Encyclopedia of Genes and Genomes (KEGG) enrichment with CKP as control is shown in Figure 3, and the *q*-value represents the corrected *p*-value, taking values between 0 and 1. It can be observed that the KEGG enrichment of DEGs is significant only in the ascorbate and aldarate metabolism pathway, and the NP group can be enriched to 11 DEGs in this pathway, with 5 DEGs upregulated and 6 DEGs downregulated. The significant enrichment of this pathway suggests that this pathway is involved in response to Pi deficiency. Other pathways, although not significant, were also enriched to a certain number of DEGs, and we can also have some concerns, such as plant hormone signal transduction, oxidative phosphorylation, and pentose phosphate pathway, which have been previously reported to be involved in P closely related to, as shown in Appendix A.

### 2.4. Screening of DEGs under Different Ethylene Treatments

#### Identification of DEGs in Four Treatment Groups

CKP_ETH (P control and ethephon treatment) vsCKP and NP_ETHvsNP groups, ethephon-imposed DGEs were available. CKP_Co (P control and CoCl_2_ treatment) vsCKP and NP_CovsNP groups, CoCl_2_-imposed DGEs were accessible. Significantly more regulatory DEGs were available in the ethephon-involved group than in the CoCl_2_-involved group, and the number of DEGs increased with increasing ethylene content Figure 4. There are 8 DEGs involved in expression in all groups (Appendix A). A total of 108 TFs were obtained in such a comparison, and 56 were significantly expressed and mostly downregulated, accounting for 67.86% (Appendix A), of which three ethylene response factor (ERF) genes were found to be significantly expressed in the CKP_ETHvsCKP group. A total of 134 genes with co-response were obtained in the CKP_ETHvsCKP and NP_ETHvsNP groups, and 17 genes with simultaneous response in the NP_ETHvsNP and NP_CovsNP groups.

### 2.5. Differential Gene Cluster Analysis in Different Treatment Groups

We performed differential gene cluster analysis on all treatment groups, and the clustering of genes in each group has its specificity. The genes of NP_ETH and NP_Co clustered clearly, and the other treatment groups clustered relatively scattered. P stress was expressed near the application of CoCl_2_, and in the Figure 5, these combinations are always clustered in adjacent branches.

## 3. Discussion

The concentration of inorganic P that soils can provide to plants is often insufficient, and P as a non-renewable resource makes us make plants pay closer attention to its efficient use [30]. This improves the use of Pi elements by plants, it is essential to understand the mechanisms of interactions between Pi and ethylene.

In our study, in addition to the study of the Chinese root transcriptome, we also did some leaf physiological indicators for chlorophyll and anthocyanin, respectively. In some groups, the differences in P response were significant, but in most of the groups, they were not significant, and the correlation with ethylene was not significant. Moreover, in the significant group, Pi starvation caused a decrease in anthocyanins differing from the results of previous studies [20]. In a recent study, the common physiological responses of Pi deficiency and ethylene were summarized in four categories involving acidification, nutrient transporters, phosphatase, chelating agents, and other compounds, with different genes involved in the different categories [31]. Induction of Pi by ethylene alone is not sufficient, and the presence of specific signals related to P in plants is abundant [32]. Ethylene plays a certain role in the non-climacteric ripening process with complex regulation [33]. Of interest is that the most obtained TFs in this study were MYB family, and we also found that Pi stress significantly downregulated MYB44 (transcript20516/f3p0/2110) in Chinese fir. As one of the largest TF family, RhMYB108 is involved in ethylene-induced petal senescence in rose, MYB30 antagonistically regulates P uptake in *Arabidopsis* under P stress, and StMYB4 negatively regulates P transport in *Solanum tuberosum* L., suggesting that MYB family genes are inextricably linked to P and hormone regulation [27,34,35]. In this comparison, we also found RAP2.11 (transcript14311/f3p0/2353), which was shown to be associated with ethylene and K regulation in previous studies. Is its differential expression under Pi deficiency because Pi deficiency also regulates the signal for K ion changes? It has also been reported in the literature that altered K content is indeed activated under Pi deficiency [36]. Ethylene, as a plant hormone, has an important role in plant growth and development, and it is not only commonly known for its ripening effect but has been repeatedly shown to regulate the physiology and morphology of plants under Pi deficiency [37]. In Pi deficiency, it can regulate plant physiology and morphology by upregulating S-adenosyl methionine synthe-tases (SAMS), 1-aminocyclopropane-1-carboxylic acid synthase (ACS), and 1-aminocyclopropane-1-carboxylic acid oxidade (ACO) genes to increase ethylene production, ethylene, in turn, can activate the final trophic response through the CTR1-EIN2-dependent pathway in response to related genes, such as ERF70 [30,38]. In this study, a total of 173 DEGs were common to NPvsCKP and NP_ETHvsCKP (Figure 2B), with genes related to Pi metabolism present, Malate dehydrogenase (transcript47986/f3p0/1238), and cytochrome b6 (transcript8729/f2p0/2658), and also a member of the ACS family mentioned above (transcript30715/f2p0/1821). Several families of TFs were also found, but some genes such as WRKY70, bHLH32, and EIN3/EIL1 (ERF family), which have been reported by ethylene deficiency on Pi, were not found, but some other members of the AP2/ERF family were found, and whether these genes have regulatory roles under Pi deficiency and exogenous ethylene application needs to be studied in depth subsequently. transcript44057/f3p0/1403 was significantly expressed in each of the ethylene-related treatments, but it was not annotated in the various databases, and we speculate that it is closely related to ethylene. Ethylene response factors belong to a large family of transcription factors, and many of its members can respond to plant ethylene signals. In soybeans, ERF1 is repressed by ethylene signaling, leading to inhibition of growth and development [39], and ERF113 was found to be responsive to ethylene in Arabidopsis [40]. In the present study, CKP_ETHvsCKP, we identified three ERF family genes, ERF113 (transcript31085/f2p0/1829), ERF1 (transcript44453/f2p0/1421), and DREB3 (transcript47328/f2p0/1291) suggesting that exogenous ethylene may have a role in the regulation of endogenous ethylene in Chinese fir and can mobilize the expression of related genes.

Research on Chinese fir, a gymnosperm, is still very limited. Although there have been many previous studies on P stress in Chinese fir, it is the first time that P stress was combined with exogenous ethylene. P stress requires strict control of the inorganic Pi concentration, but different studies have defined low P and high P differently. We refer to two more common concentrations, 1 mmol·L^−1^ KH_2_PO_4_ and 0 mmol·L^−1^ KH_2_PO_4_, and chose the Pi starvation treatment because we were concerned that the Pi concentration of the treatment was not low enough to achieve stress [7,41,42]. The choice of exogenous ethylene and ethylene inhibitors was based on a previous study on the physiology of Chinese fir, and the specific concentrations were based on similar studies in many other plants [17,43,44,45]. However, there is a wide variety of exogenous ethylene and ethylene inhibitors, and different types of compounds may yield different results, which will have to be explored later. The different times of treatment of Chinese fir seedlings and the different criteria for selecting differential genes may lead to a great difference in the number of screened differential genes. In the present study, we used *p*-adjust as the criterion for selecting differential genes, while others used *p*-value in previous studies. Such criterion can improve the accuracy of our gene screening and is beneficial for the subsequent in-depth study of a specific gene. The regulation of plant P metabolism also needs to be considered in conjunction with the cellular level, and studies relying on the transcriptome alone are insufficient. The type of RNA-seq we performed in the Illumina platform was UMI (Digital RNA-seq), which has higher accuracy compared with other types of RNA-seq. It can add a unique digital label to each cDNA fragment before reverse transcription to ensure the consistent origin of sequenced fragments and eliminate the error of PCR amplification, making the results more accurate [46]. We have not found it being used in previous studies on the transcriptome in Chinese fir.

Currently, the whole genome data of Chinese fir has not been reported, and our research is still very limited. We can only assemble a full-length transcriptome through the PacBio platform as a reference. Once the genome is available, we will be able to more comprehensively identify the mechanisms involved in the regulation of ethylene under Pi deficiency and identify the downstream target genes of a specific gene of interest using DAP-seq, for example.

## 4. Materials and Methods

### 4.1. Plant Materials, Growth Conditions, and Sampling

Experimental materials were used for 11-month-old light substrate seedlings of Yang061 clone from the state-owned forest farm of Yangkou (26°49′ N, 117°54′ E), Fujian Province, China. The Chinese fir seedlings were originally substrate washed, planted in washed river sand, and placed in the greenhouse of the Forestry College (26°04′ N, 119°14′ E), Fujian Agricultural and Forestry University. At an average growth temperature of 25 °C. The seedlings were incubated with 1/2 complete Hoagland nutrient solution for 45 days for slowing treatment, and then selected seedlings with similar growth potential were randomly divided into six groups, and the base nutrient solution for each experiment was 1/2 concentration of phosphorus-free Hoagland nutrient solution, and other added components were as follows: (1) CKP, 1 mmol·L^−^^1^ KH_2_PO_4_; (2) NP, 0 mmol·L^−1^ KH_2_PO_4_; (3) CKP_ETH, 1 mmol·L^−1^ KH_2_PO_4_ and 200 mg·L^−1^ ethephon; (4) NP_ETH, 0 mmol·L^−1^ KH_2_PO_4_ and 200 mg L^−1^ ethephon; (5) CKP_Co, 1 mmol·L^−1^ KH_2_PO_4_ and 50 μmol·L^−1^ CoCl_2_; (6) NP_Co, 0 mmol·L^−1^ KH_2_PO_4_ and 50 μmol·L^−1^ CoCl_2_, and make up the K+ concentration of Pi starvation groups with potassium chloride (KCl). Four biological replicates were performed, and a total of 24 seedlings were treated. The root was collected on day 7 of culture, and these tissues were frozen in liquid nitrogen for at least 15 min before being stored in a −80 °C refrigerator for later experiments.

### 4.2. Determination of Chlorophyll and Anthocyanin in Leaves of Chinese Fir Seedlings

Twenty-four treated seedlings were selected, and a sufficient amount of leaves were taken to determine chlorophyll and anthocyanin contents. The kits were from Shanghai Enzyme Link Biotechnology Co., Ltd. (Shanghai, China), for anthocyanin (ml076319) and chlorophyll (ml036299). Then measured by using an ELISA reader (Multiskan MS, LabSystems 325, Helsinki, Finland).

### 4.3. UMI mRNA Sequencing

Total RNA was extracted using the RNAprep Pure Plant Plus Kit (polysaccharides and polyphenolics-rich) (Tiangen, Beijing, China). The purity and concentration of RNA were measured using a NanoPhotometer spectrophotometer (NanoDrop Technologies, Wilmington, DE, USA). The integrity and quantity of RNA were assessed using an Agettent 5400 Bioanalyzer (Agilent Technologies, Santa Clara, CA, USA). Next, library preparation was performed. Total RNA was input, and mRNA was captured by mRNA capture beads with oligo (DT), purified in buffer to a random fragment of 100–200 nt, and reverse transcribed into cDNA subsequently. Ligating purified cDNA to premixed adaptors with UMI using ligase and ligase buffer was followed by one-step polymerase chain reaction. After purification, the PCR amplification products were recovered. The library was sequenced on the Illumina noveseq 6000 Platform to obtain 6G of raw data, yielding 150 nt of pair-end reads.

### 4.4. Full-Length Reference Transcripts

The material for the full-length reference transcript is a mixed sample of roots, stems, and leaves from the same Yang 061 clone of Chinese fir. Pacific Biosciences (Pacific Biosciences, Menlo Park, CA, USA) provided the official protocol for library preparation and sequence analysis. The software SMRTlink v8.0 was used to filter the downstream data. First, the CCS (circular consensus sequence) was obtained by the CCS algorithm, which sequences a single molecule multiple times and self-corrects the errors. Next, the CCS was sorted to find FLNC (full-length non-chimera) sequences and nFL (non-full-length) sequences. The FLNC sequences of the same transcript were clustered, and then the full-length sequences obtained were polished to obtain polished consensus sequences. The polished consensus data were corrected against Illumina Readers using LoRDEC software. Transcripts were finally obtained by comparative clustering of nucleic acids using CD-HIT software to remove redundant, similar sequences.

### 4.5. Functional and Structural Annotation of Unigenes

Functionally, we used NR (Non-Redundant Protein Sequence Database), NT (NCBI non-redundant nucleotide sequences), KOG/COG (Clusters of Orthologous Groups of proteins), and the BLAST tool of Swiss-Prot (a manually annotated and reviewed protein sequence database) to annotate the above full-length reference unigenes with E-value = 1 × 10^−10^, these tools can cluster and match similar genes to Chinese fir and homologous species. In addition, structural domain annotation was performed by Pfam, and pathway analysis and functional classification were performed by KEGG and GO, respectively. Structurally, we used ANGEL software for CDS prediction, followed by iTAK software for transcription factor (TF) prediction, CNCI, PLEK, CPC2 software, and Pfam-scan database were used for coding potential prediction of PacBio sequencing data, and removal of these transcripts with coding potential led to LncRNAs.

### 4.6. Transcript Differential Expression Analysis

Clean reads from 24 Illumina sequencing samples were mapped to the de-redundant full-length reference transcripts, i.e., unigenes, using RSEM software. The read count was counted after comparison, and then FPKM conversion was performed to quantify their expression. The number of DEGs for different treatments was calculated using DESeq2 with the criteria of *p*-adjust < 0.05 and |log2(FoldChange)| > 0. When |log2(FoldChange)| > 0 > 1, it is considered that the differential gene was significantly expressed [47].

### 4.7. Differential Gene Enrichment Analysis

GO enrichment was achieved by the GOseq R package, and *p*-value < 0.01 was the criterion for GO enrichment of DEGs [48]. KEGG enrichment was achieved by KOBAS (3.0), and the enriched DEGs were labeled with corrected *p*-values, and when the *q*-value was <0.05, it indicated that the differential genes were significantly enriched in this pathway. The top 20 items are reserved for both types of enrichment, and if less than 20 items are displayed.

### 4.8. Data Statistics

Physiological indicators were analyzed by ANOVA using EXCEL 2019 and SPSS 25 software. Results were expressed as “mean ± standard error” of four biological replicates and three technical replicates.

## 5. Conclusions

The DEGs were screened by six treatments of the Yang 061 clone of Chinese fir under P treatment with ethylene or CoCl_2_ treatment, and 83 TFs were obtained from them. And the MYB family was the most abundant. Using normal Pi (CKP) as a control to study the co-expression of P and ethylene, we obtained 35 co-expressed differentially expressed genes (DEGs). In these DEGs, we detected significant expression of ABC transporter, oxidation-reduction process-related genes. Ethylene responsive genes were also significantly expressed in Chinese fir with only exogenous ethylene as a variable, and we identified some new responsive genes that were not annotated. The present study differs from previous studies in that exogenous ethylene was applied in the presence of Pi deficiency to study the characteristics of RNA-Seq in Chinese fir roots, which can provide a new theoretical basis for further understanding of the response of Chinese fir to Pi deficiency.

## Figures and Tables

**Figure 1 plants-11-02036-f001:**
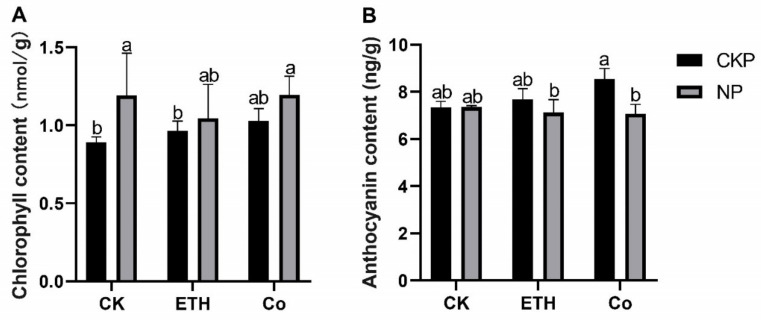
*Y*-axis represents content, and *X*-axis represents each treatment group. Gray is normal phosphate (Pi), black is Pi starvation, CK represents no additional addition, ETH represents the addition of ethephon under phosphorus (P) treatment, Co represents the addition of CoCl_2_ under P treatment. Small letters indicate significant differences between the different additives under phosphorus stress. (**A**) Chlorophyll content of Chinese fir seedlings under P and exogenous ethylene treatment. (**B**) Anthocyanin content of Chinese fir seedlings under P and exogenous ethylene treatment.

**Figure 2 plants-11-02036-f002:**
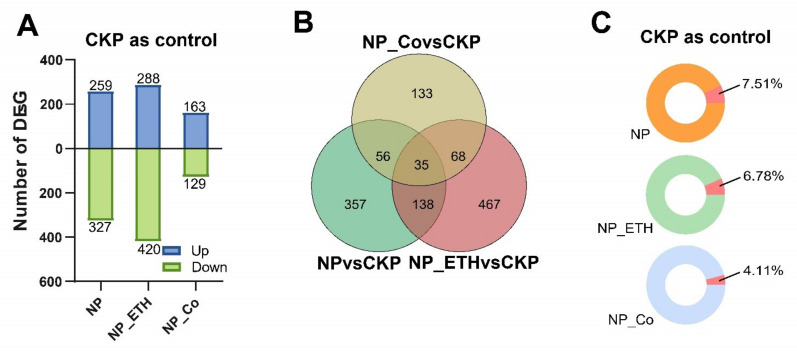
Identification of DEGs in three treatment groups, CKP (normal phosphate (Pi)) as control, NP (Pi starvation), NP_ETH (Pi starvation and ethephon) and NP_Co (Pi starvation and CoCl_2_) were each experimental group. (**A**) The histogram indicates upregulated and downregulated DEGs, blue indicates up-regulation, green indicates down-regulation, *Y*-axis indicates the number of DEGs, and *X*-axis represents the experimental group name. (**B**) Venn diagram of the DEGs among three treatment groups, and the number represents the number of DEGs. (**C**) Donut chart indicates the percentage of TFs in three groups. NP (Pi starvation treatment), NP_ETH (Pi starvation and ethephon treatment), NP_Co (Pi starvation and CoCl_2_ treatment).

**Figure 3 plants-11-02036-f003:**
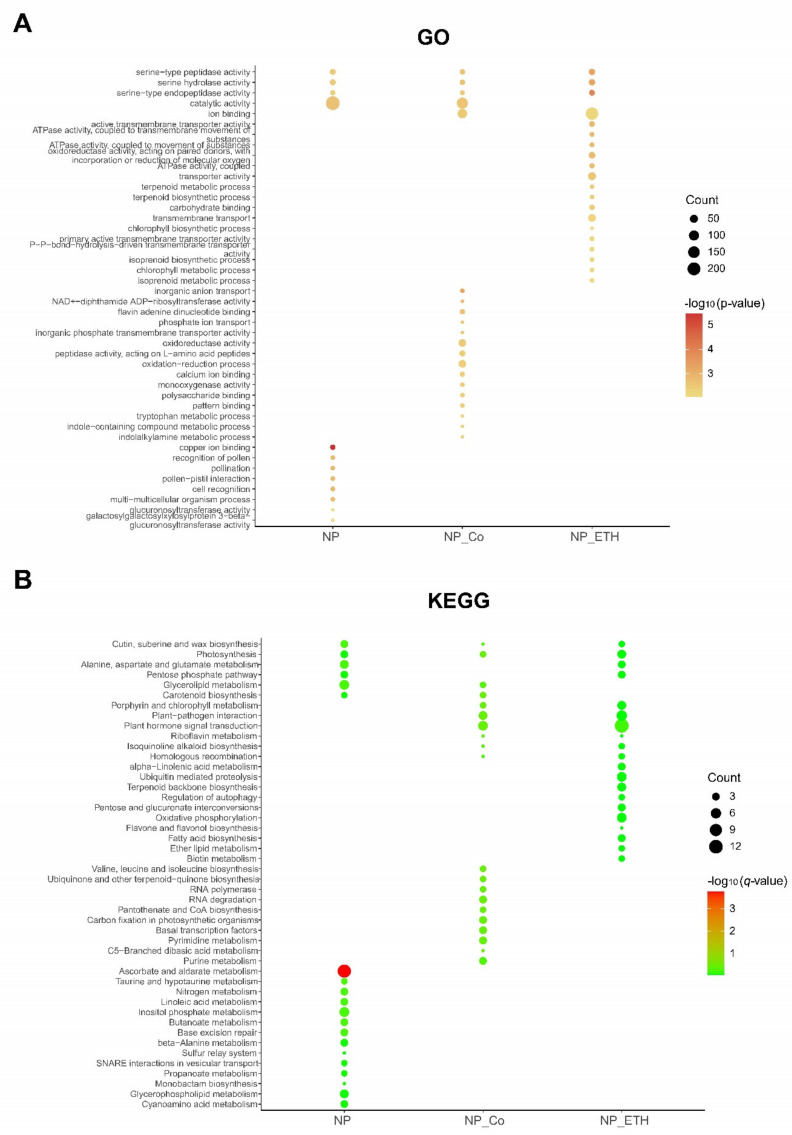
The bubble diagram shows that the DEGs of (**A**) GO and (**B**) KEGG were under three groups of treatment., Y-axis is the description of the function, and X-axis is the group name. CKP (normal Pi) as control, NP (Pi starvation), NP_ETH (Pi starvation and ethephon), NP_Co (Pi starvation and CoCl_2_) were each experimental group point size variable represents the number of enriched genes. Color variables correspond to different −log10(*q*-value) or −log10(*p*-value) ranges.

**Figure 4 plants-11-02036-f004:**
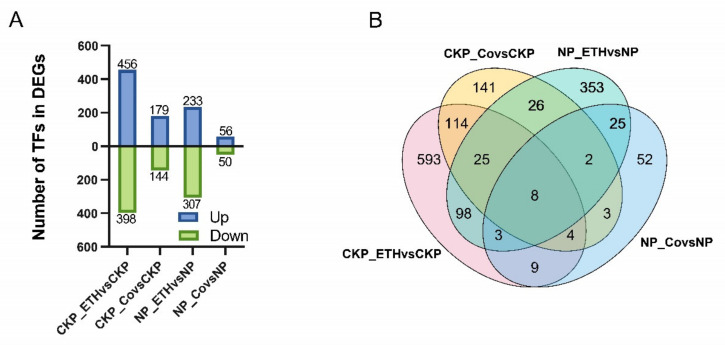
Identification of DEGs in four treatment groups, namely CKP (normal phosphate (Pi), NP (Pi starvation), CKP_ETH (normal Pi and ethephon), CKP_Co (normal Pi and CoCl_2_), NP_ETH (Pi starvation and ethephon), NP_Co (Pi starvation and CoCl_2_). (**A**) The histogram indicates upregulated and downregulated DEGs, blue indicates up-regulation, green indicates down-regulation, *Y*-axis indicates the number of DEGs, and *X*-axis represents the experimental group name. (**B**) Venn diagram of the DEGs among four treatment groups, and the number represents the number of DEGs.

**Figure 5 plants-11-02036-f005:**
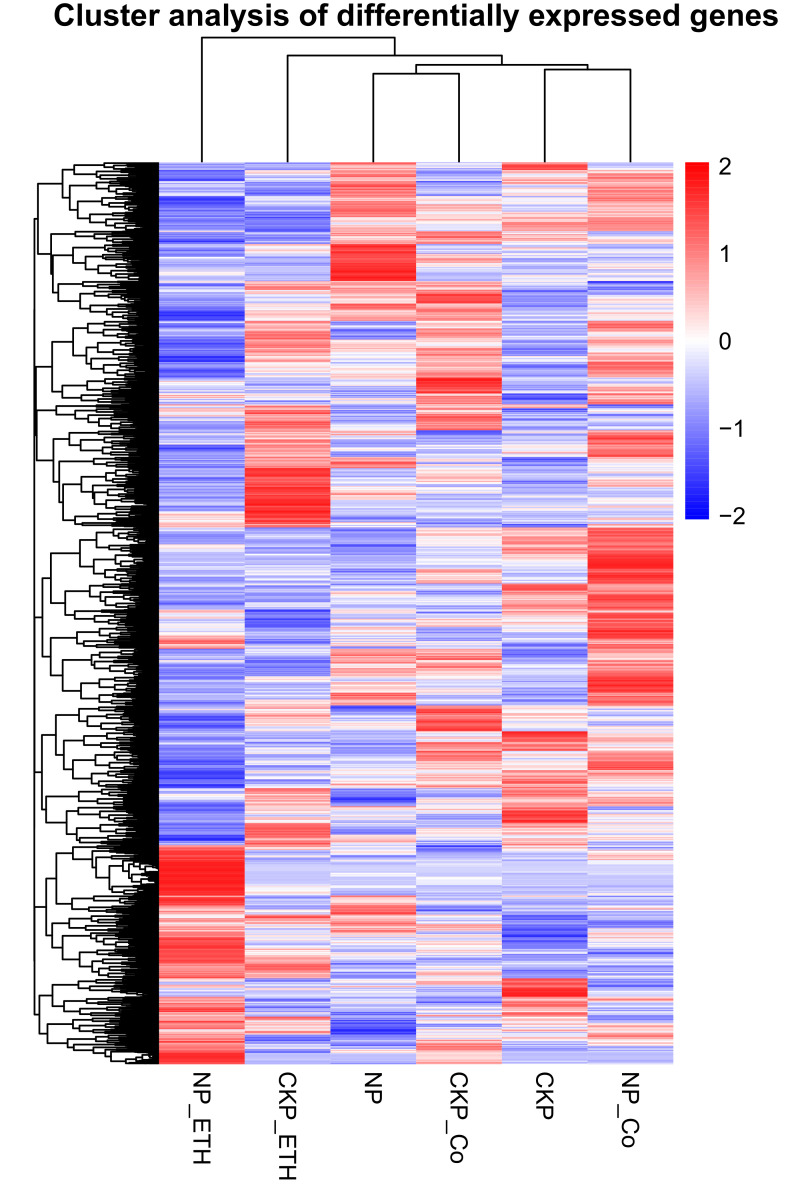
Gene cluster analysis among different treatments, namely CKP (normal phosphate (Pi)), NP (Pi starvation), CKP_ETH (normal Pi and ethephon), CKP_Co (normal Pi and CoCl_2_), NP_ETH (Pi starvation and ethephon), NP_Co (Pi starvation and CoCl_2_). The abscissa represents the group name, and the color represents the degree of expression. Red indicates high expression genes, blue indicates low expression genes, and red to blue indicates log10(FPKM+1) from large to small.

**Table 1 plants-11-02036-t001:** Overview of Chinese fir PacBio platform sequencing.

Name	Reads Number	Average Length
Polymerase reads	698,758	106,051
Subreads	36,840,729	1936
CCS	631,560	2110
FLNC	564,893	1966
Consensus	53,413	2015
Unigenes	22,243	2079

**Table 2 plants-11-02036-t002:** Significant of DEGs under P treatment with the addition of ethephon or CoCl_2_.

Gene_id	NPvsCKP	NP_ETHvsCKP	NP_CovsCKP
transcript11003/f13p0/2395	−2.12	−4.2817	−2.3369
transcript21015/f3p0/2069	−2.6494	−4.2227	−1.75
transcript13260/f5p0/2398	−2.6975	−6.5059	1.0092

All genes in the table *p*-adjust < 0.05, and the values in the table represent log2foldchange.

## Data Availability

All raw data reported in this article have been deposited in the National Center. The names of the accession number(s) can be found below: PRJNA850073 and PRJNA850136.

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
