# Peer review of "Transcriptome Level Analysis of Genes of Exogenous Ethylene Applied under Phosphorus Stress in Chinese Fir"

_plants, 2022, doi:10.3390/plants11152036_

Round 1
Reviewer 1 Report
This is an interesting paper, but you may improve this article to publish in this journal. Otherwise, I have a lot of recommendations to increase the quality of your paper. Be careful with the writing and mistakes.
In the Abstract there are some mistakes.
Line 21. When you write an abstract you must write the meaning of every acronym because the abstract is the first contact with potential readers and is in public platforms. So, you must write the meaning of “DEGs”. Fix this very common mistake in all your document.
Line 22. Just after the number 292 you must write the word “respectively”.
Line 23. Please, write the meaning of the acronyms “MYB” and “ERF”.
There are two keywords repeated in the article title. The keyword are “Chinese fir” and ”ethylene”. In order to increase the visibility of your paper I recommend changing these keywords. If you change it by other keywords, you will increase the probability that your paper could be found by future readers when they look for your paper in some databases like Scopus for example. If you repeat the same words in the article title and in keywords, less people could find your work. So, you must think about the visibility of your research.
Line 32. When you write a reference in your manuscript you must write a space just before the square brackets. So, you must write as follows: “…properties [1].”. This is a very common mistake in your paper, and you must fix it in the whole document.
Line 38. When you write two references inside the square brackets you must delete the space between the numbers. When you write an article, you must read at least an article of the journal where you are submitting a paper in order to copy the stile. So, you must write “[3,4].”. Please, fix this mistake in your whole document.
Line 42. Every time you write an acronym you must write the letters in capitals. This make the reading easier. So, you must write “Pi Starvation Response (PSR)”. Please, fix this mistake in all your paper.
Line 48. Remember, every time that you write an acronym you must write its meaning. So, write in brackets the meaning of “ABA”. You must fix this very common mistake in all your manuscript.
Line 60. Just after the point you must write the words in capitals, this is very easy and basic grammar. So, you must write the word “ethylene” in capitals just after the point.
Line 66. Just because this is a botanical journal you must write the author just after a scientific name at least the first time that you write it in your article. You must fix it in your whole paper.
You must write the objectives of your investigation explicitly.
Line 103. Remember that every time that you write an acronym you must write its meaning. So, you must write the meaning of “CKP”. I insist that you must read your whole document and look for this mistake and fix it.
Line 103-108. When you write the figure (Figure 1A), you must put it into brackets, this make the reading easier.
Line 110. Just after a point you must write the words in capitals, this is basic grammar. So, you must write as follows: “…to Pi. The TFs…”. Please, read one more time your document and look for this mistake in your whole work.
Line 123. You must write “Table 2” into the brackets. Fix this mistake in your manuscript.
Line 129. When you write an article, you must think first in the reader because your are trying to communicate science, so, please, write the meaning of every acronym at least the first time that you write it. So, you must write the meaning of the acronym “GO”.
Line 131. Please, write the letters that you use for the acronym in capitals. This makes the reading easier. So, for example you must write “Molecular Function (MF)”. Fix this mistake in all your manuscript.
Line 137. Please, write the meaning of the acronym “KEGG”.
Line 145. You must write in capitals the words just after a point, not after commas. This is basic grammar, so you must write “…transduction, oxidative phosphorylation…”.
Line 157. Just before the bracket you must write a space. So, you must write as follows: “…in all groups (Table S6).”.
Line 164. It is very difficult to read the letters of the figures because they are very small. Please, fix it.
Line 171-174. I cannot read the Figure 4 in the text, so, you must refer to it.
Line 175. It is impossible to read anything. So, you must make figures 3 and 4 bigger.
Line 188. The element phosphorus must be written with a “P” in capitals. This basic chemistry. You must fix this mistake in all your manuscript in lines 194, 215 and 231 as well, and you must look for this mistake in your whole work.
Line 193. You must write the scientific names in italics.
Line 194. You must write “potato” using scientific names because this is a botanical journal.
Line 247. You must provide a map of the distribution of the samples with the distribution of the species as well.
Line 286. You must write the letters of the acronym in capitals.
Otherwise, the authors adequately developed the Introduction, presenting the problems but you must write explicitly the objectives of this paper.
The methods are adequate.
The Discussion is well developed, and the data presented are correctly compared with other papers.
The authors are to be congratulated for the results obtained in this article.
Author Response
Response to Reviewer 1 Comments
Dear reviewer:
Thank you for your decision and constructive comments on my manuscript. This response is based on the file "plants-1810998-1-Revised version". The red part has been revised according to your comments. Revision notes, point-to-point, are given as follows:
Point 1: Line 21. When you write an abstract you must write the meaning of every acronym because the abstract is the first contact with potential readers and is in public platforms. So, you must write the meaning of “DEGs”. Fix this very common mistake in all your document.
Response 1: We are very sorry for our incorrect writing and it is rectified at Line 22.
Point 2: Line 22. Just after the number 292 you must write the word “respectively”.
Response 2: We are very sorry for our incorrect writing and it is rectified at Line 23.
Point 3: Line 23. Please, write the meaning of the acronyms “MYB” and “ERF”.
Response 3: We are very sorry for our incorrect writing and it is rectified at Line 24 and Line 26.
Point 4: There are two keywords repeated in the article title. The keyword are “Chinese fir” and ”ethylene”. In order to increase the visibility of your paper I recommend changing these keywords. If you change it by other keywords, you will increase the probability that your paper could be found by future readers when they look for your paper in some databases like Scopus for example. If you repeat the same words in the article title and in keywords, less people could find your work. So, you must think about the visibility of your research.
Response 4: We agree with this recommendation and the keywords are replaced in Line 31 and Line32.
Point 5: Line 32. When you write a reference in your manuscript you must write a space just before the square brackets. So, you must write as follows: “…properties [1].”. This is a very common mistake in your paper, and you must fix it in the whole document.
Response 5: We are very sorry for this error, and the formatting has been corrected in the full manuscript.
Point 6: Line 38. When you write two references inside the square brackets you must delete the space between the numbers. When you write an article, you must read at least an article of the journal where you are submitting a paper in order to copy the stile. So, you must write “[3,4].”. Please, fix this mistake in your whole document.
Response 6: We are very sorry for this error, and the formatting has been corrected in the full manuscript.
Point 7: Line 42. Every time you write an acronym you must write the letters in capitals. This make the reading easier. So, you must write “Pi Starvation Response (PSR)”. Please, fix this mistake in all your paper.
Response 7: We are very sorry for our incorrect writing and it is rectified at Line 48 and checked the full manuscript.
Point 8: Line 48. Remember, every time that you write an acronym you must write its meaning. So, write in brackets the meaning of “ABA”. You must fix this very common mistake in all your manuscript.
Response 8: We are very sorry for our incorrect writing and it is rectified at Line 54.
Point 9: Line 60. Just after the point you must write the words in capitals, this is very easy and basic grammar. So, you must write the word “ethylene” in capitals just after the point.
Response 9: We are very sorry for our mistake, this part has been replaced with other content in Line 82-84.
Point 10: Line 66. Just because this is a botanical journal you must write the author just after a scientific name at least the first time that you write it in your article. You must fix it in your whole paper.
Response 10: We are very sorry for our incorrect writing and it is rectified at Line 61.
Point 11: You must write the objectives of your investigation explicitly.
Response 11: Thanks to the reviewers' suggestions, we have added this section in Line 106-110.
Point 12: Line 103. Remember that every time that you write an acronym you must write its meaning. So, you must write the meaning of “CKP”. I insist that you must read your whole document and look for this mistake and fix it.
Response 12: We are very sorry for our incorrect writing and it is rectified at Line 149 and checked the full manuscript.
Point 13: Line 103-108. When you write the figure (Figure 1A), you must put it into brackets, this make the reading easier.
Response 13: We are very sorry for our incorrect writing and it is rectified at Line 154.
Point 14: Line 110. Just after a point you must write the words in capitals, this is basic grammar. So, you must write as follows: “…to Pi. The TFs…”. Please, read one more time your document and look for this mistake in your whole work.
Response 14: We are very sorry for our incorrect writing and it is rectified at Line 158 and checked the full manuscript.
Point 15: Line 123. You must write “Table 2” into the brackets. Fix this mistake in your manuscript.
Response 15: We are very sorry for our incorrect writing and it is rectified at Line 174.
Point 16: Line 129. When you write an article, you must think first in the reader because your are trying to communicate science, so, please, write the meaning of every acronym at least the first time that you write it. So, you must write the meaning of the acronym “GO”.
Response 16: We are very sorry for our incorrect writing and it is rectified at Line 181.
Point 17: Line 131. Please, write the letters that you use for the acronym in capitals. This makes the reading easier. So, for example you must write “Molecular Function (MF)”. Fix this mistake in all your manuscript.
Response 17: We are very sorry for our incorrect writing and it is rectified at Line 181-183.
Point 18: Line 137. Please, write the meaning of the acronym “KEGG”.
Response 18: We are very sorry for our incorrect writing and it is rectified at Line 190.
Point 19: Line 145. You must write in capitals the words just after a point, not after commas. This is basic grammar, so you must write “…transduction, oxidative phosphorylation…”.
Response 19: We are very sorry for our incorrect writing and it is rectified at Line 198.
Point 20: Line 157. Just before the bracket you must write a space. So, you must write as follows: “…in all groups (Table S6).”.
Response 20: We are very sorry for our incorrect writing and it is rectified at Line 214.
Point 21: Line 164. It is very difficult to read the letters of the figures because they are very small. Please, fix it.
Response 21: We replaced the figure with a new style in Line 221.
Point 22: Line 171-174. I cannot read the Figure 4 in the text, so, you must refer to it.
Response 22: Figure 4 has been renamed Figure 5, and we have added more detailed notes to Figure 5 in Line 235-240.
Point 23: Line 175. It is impossible to read anything. So, you must make figures 3 and 4 bigger.
Response 23: Figure 3 has been renamed Figure 4, and We replace Figure 4 and Figure 5 with a larger look in Line 222 and Line 229.
Point 24: Line 188. The element phosphorus must be written with a “P” in capitals. This basic chemistry. You must fix this mistake in all your manuscript in lines 194, 215 and 231 as well, and you must look for this mistake in your whole work.
Response 24: We are very sorry for our incorrect writing and it is rectified in Line 243, 256, 262 and 283.
Point 25: Line 193. You must write the scientific names in italics.
Response 25: We are very sorry for our incorrect writing and it is rectified at Line 261.
Point 26: Line 194. You must write “potato” using scientific names because this is a botanical journal.
Response 26: We are very sorry for our incorrect writing and it is rectified in Line 262-263.
Point 27: Line 247. You must provide a map of the distribution of the samples with the distribution of the species as well.
Response 27: We added the sample distribution with the sample planting map in Line 330, 332 and 333.
Point 28: Line 286. You must write the letters of the acronym in capitals.
Response 28: We are very sorry for our incorrect writing and it is rectified at Line 372.
Point 29: Otherwise, the authors adequately developed the Introduction, presenting the problems but you must write explicitly the objectives of this paper.
Response 29: Thanks to the reviewers' suggestions, we have added this section in Line 106-110.
Thanks again to the reviewer for giving valuable suggestions.

Reviewer 2 Report
In this manuscript, authors carried out transcriptome analysis of Chinese fir under normal and phosphorous starving conditions together with exogenous ethylene treatment. In my understanding, most of the data is preliminary and many speculative inferences are made without providing sufficient evidence. My specific comments are below:
1. The study is planned poorly and lacks a clear hypothesis. For instance, what is CoCl2 and why it is included in the treatment group, it is not clear. Authors should provide the rationale for this.
2. What is basis of selecting different concentrations for P stress, ethylene and CoCl2 treatment? Whether authors performed any screening to select these conditions?
3. What was the phenotype of the plants after different treatment? Please include this data in the manuscript.
4. I would suggest analyzing the Pi stress marker genes and ethylene biosynthesis and signaling genes under different treatments and provide a new figure for this. Further, the expression of the above genes should be validated with qRT-PCR.
5. Discussion is weakly written and should be improved with proper references to justify the statements.
6. Abbreviations must be defined at their first mention in the manuscript including the abstract.
7. There are several typographical errors throughout the manuscript.
8. There are many places where the author's message is not clear due to improper phrasing and English language. Authors need to check that throughout the manuscript.
Author Response
Response to Reviewer 2 Comments
Dear reviewer:
Thank you for your decision and constructive comments on my manuscript. This response is based on the file "plants-1810998-1-Revised version". The red part has been revised according to your comments. Revision notes, point-to-point, are given as follows:
Point 1: The study is planned poorly and lacks a clear hypothesis. For instance, what is CoCl2 and why it is included in the treatment group, it is not clear. Authors should provide the rationale for this.
Response 1: We thank the reviewers for pointing this out. We should indeed add this aspect, as detailed in Line 69-72 and Line 106-110.
Point 2: What is basis of selecting different concentrations for P stress, ethylene and CoCl2 treatment? Whether authors performed any screening to select these conditions?
Response 2: Choosing different concentrations for phosphorus stress, ethylene, and CoCl2 treatments we consulted the papers of Chinese fir and other species related studies [1-5]. Because the boundaries of exactly how low phosphorus are unclear, we chose to treat with phosphorus starvation, which is also frequently used in much of the literature. 200 mg/L ethephon, 5 μmol/L CoCl2 were referenced from a Chinese master's thesis on Triticum aestivum L. and a paper on Chinese fir [6,7]. These contents are also added to the Line 296-302.
Point 3: What was the phenotype of the plants after different treatment? Please include this data in the manuscript.
Response 3: We supplemented some physiological indicators, chlorophyll and anthocyanin content, in Line 116-129.
Point 4: I would suggest analyzing the Pi stress marker genes and ethylene biosynthesis and signaling genes under different treatments and provide a new figure for this. Further, the expression of the above genes should be validated with qRT-PCR.
Response 4: We thank the reviewers for the suggestions given. Pi stress marker genes under different treatments and ethylene biosynthesis and signaling genes in Line147-179. We believe that qRT-PCR experiments to verify transcriptome expression are very important, but due to the limited time to change the manuscript, we were not able to supplement the qRT-PCR experiments, and we apologize for this. However, the type of RNA-seq we performed on Illumina platform is UMI (digital RNA-seq), which can add a unique digital tag to each cDNA fragment before reverse transcription to ensure the consistent origin of sequenced fragments and eliminate PCR amplification errors, which will make our results more accurate. In addition, our criteria for screening differential genes is the corrected P value, which further enhances the accuracy of the results.
Point 5: Discussion is weakly written and should be improved with proper references to justify the statements.
Response 5:We have added to the discussion in Line 247-252 and Line 296-309.
Point 6: Abbreviations must be defined at their first mention in the manuscript including the abstract.
Response 6: We are very sorry for our incorrect writing and it is rectified in Line 13-30 and checked the full manuscript.
Point 7: There are several typographical errors throughout the manuscript.
Response 7: We are very sorry for our writing errors and have checked the full manuscript.
Point 8: There are many places where the author's message is not clear due to improper phrasing and English language. Authors need to check that throughout the manuscript.
Response 8: We are very sorry for our writing errors and have checked the full manuscript.
Thanks again to the reviewer for giving valuable suggestions.
- Chen, R.; Zhou, M.; Li, J.; Zhong, W.; Wu, P.; Ma, X.; Li, M. Effects of sucrose addition on response of Chinese fir to low phosphorus stress and sucrose metabolism. Acta Ecologica Sinica 2021, 41 (16), 6588-6599.
- Gao, B.; Cao, C. L.; Tao, L. I. Effect of Ethylene on Morphology and Physiological Characteristic of Soybean Seedlings under Low-Phosphorus Stress. Soybean Science 2012.
- Sun, T.; Zhang, J.; Zhang, Q.; Li, X.; Li, M.; Yang, Y.; Zhou, J.; Wei, Q.; Zhou, B. Transcriptome and metabolome analyses revealed the response mechanism of apple to different phosphorus stresses. Plant Physiol Biochem 2021, 167, 639-650.
- Zhang, H.; Wang, N.; Zheng, S.; Chen, M.; Ma, X.; Wu, P. Effects of Exogenous Ethylene and Cobalt Chloride on Root Growth of Chinese Fir Seedlings under Phosphorus-Deficient Conditions. Forests 2021, 12 (11).
- Chen, W.; Zhou, M.; Zhao, M.; Chen, R.; Tigabu, M.; Wu, P.; Li, M.; Ma, X. Transcriptome analysis provides insights into the root response of Chinese fir to phosphorus deficiency. BMC Plant Biol 2021, 21 (1), 525.
- Li, J. Effect of ethylene on root morphology&astomataand carbohydrate content in Triticum aestivum L. under phosphate stress. Master, Northwest A&F University, China, May 2012.
- Chen, M.; Lai, H.; Zheng, S.; Li, M.; Ma, X.; Wu, p. Effects of Exogenous Ethylene on Growth and Phosphorus Use Efficiency of Chinese Fir Seedlings under Phosphorus Stress. Scientia Silvae Sinicae 2021, 57.

Reviewer 3 Report
The findings presented in this work are potentially interesting. The authors provide transcriptome analysis of Chinese fir exposed to phosphorus stress and ethylene. The manuscript has to be thoroughly revised to correct the English language mistakes. In addition, I have a number of comments concerning certain statements and conclusions which, in my opinion, should be addressed in order to consider the manuscript for publication:
- The text requires a really good English language check, there are a lot of grammar and style mistakes throughout the text.
- The authors use abbreviations in the summary, such as “CoCl2”, “ISO-Seq”, “UMI RNA-Seq”etc., the meaning of which is not immediately obvious. I suggest the authors avoid using these terms here and rather define clearly the biological questions raised and main achievements of this study.
- Line 32. Clarify the sentence. Also, the connection with the following sentence is not clear. How phosphorus connects with continuous planting of Chinese fir for several generations and its material quality?
- Line 47-62. Also the authors make a summary about the link between the hormones and Pi deficiency, it would important to mention in which plant species the links where shown.
- Line 63. Correct the sentence. It is very general and some clarifications on what signaling pathways ethephon affects and in which species, are required. Why did the authors choose ethephone and CoCl2 treatments in this study? This is important as this treatments are the core of the study.
- Line 72-76. The sentence is very vague and too generalized, please clarify and elaborate.
- Line 80. “The introduction should briefly place the study in a broad context and highlight why it is important. It should define the purpose of the work and its significance.” It sounds like this sentence is taken from guidelines for authors for writing articles. The authors forgot to remove it? Please check
- The results section needs some heavy work on clarifying the abbreviations, terminology, sample names and providing the explanations of the results in a clear and logical way.
- Figure 1. Clear explanations of abbreviations are required.
- Figure 2. The figure is very confusing and the authors should provide a clear legend indicating what is shown and what are the counts indicating. Perhaps the terms on the Y axis could be grouped according to their functional similarity.
- Figure 3. Figure abbreviations should be clarified in the legend. What are the values in the Y axis? Also, the authors provide the numbers of TFs differentially regulated, it is hard to appreciate the level of their up or downregulation. At the same time, it would be a good idea to provide here examples of specific TFs potentially in ethylene and ABA signaling.
- Figure 4. Figure abbreviations should be clarified in the legend. Also, correct the mistakes in the legend. What does the colour bar show exactly? Add corresponding labels.
- The discussion is very chaotic and random. Why was so important to use ethephone and CoCl2 treatments? What does it tell us? The authors should discuss more clearly and logically about the outcome of this study and its potential impact.
Author Response
Response to Reviewer 3 Comments
Dear reviewer:
Thank you for your decision and constructive comments on my manuscript. This response is based on the file "plants-1810998-1-Revised version". The red part has been revised according to your comments. Revision notes, point-to-point, are given as follows:
Point 1: The text requires a really good English language check, there are a lot of grammar and style mistakes throughout the text.
Response 1: Thanks to the reviewer's suggestion, we checked the full manuscript for grammar and corrected the grammatical errors.
Point 2: The authors use abbreviations in the summary, such as “CoCl2”, “ISO-Seq”, “UMI RNA-Seq”etc., the meaning of which is not immediately obvious. I suggest the authors avoid using these terms here and rather define clearly the biological questions raised and main achievements of this study.
Response 2: We thank the reviewer for the comment. We removed the terms as much as possible based on the comments and annotated the ambiguous terms in Line 20, but we left the UMI RNA-Seq as this is where we feature in this experiment and it enhances the accuracy of our results, explained in Line 313-320.
Point 3: Line 32. Clarify the sentence. Also, the connection with the following sentence is not clear. How phosphorus connects with continuous planting of Chinese fir for several generations and its material quality?
Response 3: We clarified the sentence in Line 37-38.
Point 4: Line 47-62. Also the authors make a summary about the link between the hormones and Pi deficiency, it would important to mention in which plant species the links where shown.
Response 4: We linked these symptoms to specific plant species based on comments in Line 58-90.
Point 5: Line 63. Correct the sentence. It is very general and some clarifications on what signaling pathways ethephon affects and in which species, are required. Why did the authors choose ethephone and CoCl2 treatments in this study? This is important as this treatments are the core of the study.
Response 5: We modified the ambiguous sentences in the comments in Line 82-83.
Point 6: Line 72-76. The sentence is very vague and too generalized, please clarify and elaborate.
Response 6: We revised the sentences according to the comments in Line 102-110.
Point 7: Line 80. “The introduction should briefly place the study in a broad context and highlight why it is important. It should define the purpose of the work and its significance.” It sounds like this sentence is taken from guidelines for authors for writing articles. The authors forgot to remove it? Please check
Response 7: We are very sorry for our incorrect writing and it is rectified at Line 112-114.
Point 8: The results section needs some heavy work on clarifying the abbreviations, terminology, sample names and providing the explanations of the results in a clear and logical way.
Response 8: We have interpreted the abbreviations in the results based on the comments in Line 411-423.
Point 9: Figure 1. Clear explanations of abbreviations are required.
Response 9: Figure 1 has been renamed as Figure2. We have made changes based on the comments in Line 164-166.
Point 10: Figure 2. The figure is very confusing and the authors should provide a clear legend indicating what is shown and what are the counts indicating. Perhaps the terms on the Y axis could be grouped according to their functional similarity.
Response 10: Figure 2 has been renamed as Figure 3. We replaced the clear figure according to the comments and added more detailed notes in Line 202-206.
Point 11: Figure 3. Figure abbreviations should be clarified in the legend. What are the values in the Y axis? Also, the authors provide the numbers of TFs differentially regulated, it is hard to appreciate the level of their up or downregulation. At the same time, it would be a good idea to provide here examples of specific TFs potentially in ethylene and ABA signaling.
Response 11: Figure 3 has been renamed as Figure 4. We clarified the abbreviations of the graphs according to the comments in Line 164-166, and the specific data on differentially regulated TF we added in Table S4. Values in the Y-axis are representative of the number of differential genes. Our results did not identify specific TFs that may be present in ABA signaling.
Point 12: Figure 4. Figure abbreviations should be clarified in the legend. Also, correct the mistakes in the legend. What does the colour bar show exactly? Add corresponding labels.
Response 12: Figure 4 has been renamed as Figure 5. We clarify figure abbreviations based on comments and correct errors in the legend in Line 236-240.
Point 13: The discussion is very chaotic and random. Why was so important to use ethephone and CoCl2 treatments? What does it tell us? The authors should discuss more clearly and logically about the outcome of this study and its potential impact.
Response 13: We modified the discussion based on the comments to add the reasons for ethephone and CoCl2 treatment to Line 296-309.
Thanks again to the reviewer for giving valuable suggestions.

Round 2
Reviewer 2 Report
Authors have responded satisfactorily and the paper may be accepted for publication.